# Doppler-Spectrum Feature-Based Human–Vehicle Classification Scheme Using Machine Learning for an FMCW Radar Sensor

**DOI:** 10.3390/s20072001

**Published:** 2020-04-02

**Authors:** Eugin Hyun, YoungSeok Jin

**Affiliations:** Division of Automotive Technology, ICT Research Institute, Convergence Research Institute, DGIST, 333, Techno Jungang-daero 333, Hyeonpung-myeon, Dalseong-gun, Daegu 42988, Korea; ysjin@dgist.ac.kr

**Keywords:** human detection, FMCW radar, range-Doppler processing, radar machine learning

## Abstract

In this paper, we propose a Doppler-spectrum feature-based human–vehicle classification scheme for an FMCW (frequency-modulated continuous wave) radar sensor. We introduce three novel features referred to as the scattering point count, scattering point difference, and magnitude difference rate features based on the characteristics of the Doppler spectrum in two successive frames. We also use an SVM (support vector machine) and BDT (binary decision tree) for training and validation of the three aforementioned features. We measured the signals using a 24-GHz FMCW radar front-end module and a real-time data acquisition module and extracted three features from a walking human and a moving vehicle in the field. We then repeatedly measured the classification decision rate of the proposed algorithm using the SVM and BDT, finding that the average performance exceeded 99% and 96% for the walking human and the moving vehicle, respectively.

## 1. Introduction

Currently, in the commercial market, radar sensors are applied to various platforms and applications, such as vehicles, robots, and drones, as well as medical, electronic, and safety applications [1]. Specifically, this market expansion was accelerated by the radar transceiver chipsets released by major vendors, such as Infineon, TI, NXP, STMicroelectronics, and Analog Devices, to name a few. Moreover, various software tool vendors also provide a variety of radar signal processing functions to users.

Compared to other optic sensors such as cameras and LIDAR (light detection and ranging), because radar sensors are highly robust to external environmental conditions such as weather and illuminance, these sensors can be used for a wider variety of surveillance applications for object monitoring. Whereas camera sensors can identify various classes of objects given the significant advances in machine learning techniques, conventional radar sensors can only detect objects as point objects, regardless of the type of object.

Recently, research on the detection of human activities and motions was conducted. Examples include hand gesture recognition [2], human gait indication [3], human fall detection [4], and human vital signal detection [5].

In particular, research to distinguish between humans and other objects is important with regard to various smart applications. If we can recognize whether or not a detected object is a human, it is possible to support smart security, surveillance, and unmanned vehicles in various areas, such as cities, buildings, homes, and urban streets.

The conventional approach for human indication is based on micro-Doppler signatures produced by the human body. Because humans have non-rigid motion, additional patterns exist in the Doppler frequency spectrum, appearing as sidebands around the Doppler frequency of bulk motions such as human walking [6,7,8,9]. However, because we should measure the radar echo signals to analyze the patterns of the micro-Doppler images during several measurement instances, this solution cannot meet the real-time requirement. Moreover, when using this method, because a Doppler radar is mainly used, the approach cannot handle range measurements and multi-target detection processes.

In order to resolve these problems, FMCW (frequency-modulated continuous wave) radar-based target classification methods are researched. The advantage of FMCW radar is the ability to detect the range, velocity, and angle position of a target, compared to the typical CW Doppler radar.

In earlier works [10,11,12], despite the fact that an FMCW radar was employed, time-varying micro-Doppler signatures were analyzed for human activities and human–vehicle classification. Therefore, in these cases, real-time indications can be challenging because several time measurements are required.

To resolve this problem, in various researches [12,13], a feature extraction scheme based on a two-dimensional (2D) range–velocity matrix was used for pedestrian classification. In this case, because image processing using a 2D map is used, higher time complexity for signal processing may be required. Moreover, in the field, because the scattering points of the Doppler spectrum returned from a walking human are highly variable with every measurement instance, the classification performance may be limited.

As another typical FMCW based method [14], different features of the range and velocity profiles were applied to classify humans and vehicles. This paper assumed that the reflected range profile of a moving human can be sharpened, and the Doppler spectrum can be widened. On the other hand, for a moving vehicle, several range echoes can be distributed, and a point-shaped Doppler profile may arise. However, because the variation of Doppler spectrum echoes from human over every frame is not considered, the reliability of this method can be limited in the field.

In another approach [15] based on a newly defined parameter using the RCS (radar cross-section), three significant features were extracted from the received radar signal and used as classification criteria to identify humans and vehicles. This method can be operated in real time with a simple classification function. However, performance improvements of the method based on the magnitude of the echo signal are limited, because the received reflections depend on many factors, including the shape of the target, the range-angle position, and the moving direction of the target.

In other recent articles [16,17,18], an FMCW-radar-based deep-learning technique was investigated. These types of approaches recognize not only the position of a target but also the type of target. In such a case, a neural network is applied using 2D on the range-angle domain or range-Doppler domain or using three dimensions (3D) in the range-*xy* domain. In this case, because 2D or 3D radar images with high range resolutions and high angle resolutions are needed for better performance, very high computational loads are incurred, and the hardware and software burdens are increased.

Thus, in this paper, we propose a new human–vehicle classification scheme, which uses three features from the Doppler spectrum. The proposed algorithm is designed based on FMCW radar. In the proposed algorithm, target detection is carried out initially to determine the range and velocity of the object, as well as the corresponding Doppler spectrum.

In the next step, using the Doppler spectra of two successive frames oriented from the same target, we extract three features to represent the Doppler scattering characteristics of a human and a vehicle.

Because the Doppler spectrum of a walking human can be broader due to non-rigid motion, we define the first vector while counting the Doppler reflection points in the current frame. Next, because the Doppler spectrum of a human in the current frame is not always wide enough but contains much variation compared to the previous frame, the second feature is defined as the difference in the Doppler scattering points between two successive frames. Finally, we consider that the magnitude of the Doppler spectrum of the human can fluctuate more; thus, we generate the maximum magnitude variation rate as the third feature.

We previously presented the concept of two features in an earlier study [19]. However, in that study, we did not consider that the echo power of a human could vary because the Doppler spectrum of a human fluctuated more. Moreover, we only analyzed the characteristics of the Doppler scattering points of a human and a vehicle but did not apply them to machine learning. Thus, in the present paper, we define the third feature to improve the performance of human–vehicle classification. Moreover, we extract three features using the actual measurement data from a 24-GHz FMCW radar transceiver and verify the proposed human–vehicle classification scheme using a support vector machine approach and a binary decision tree.

In Section 2, we propose the human–vehicle classification scheme based on Doppler spectrum features with machine learning. In Section 3, we present the verification results using real data from a 24-GHz FMCW radar front-end module and a real-time data acquisition module. Finally, we present the conclusions of our study in Section 4.

## 2. Proposed Human Indication Scheme

In this paper, we employ a fast-ramp-based 2D range-Doppler FMCW radar technique. The concept is very effective when used to measure the range and velocity of a target simultaneously [20,21,22]. Figure 1 shows the basic concept of a fast-ramp-based FMCW radar with a saw-tooth unchangeable for all ramps.

From the beat signals received in every ramp, 2D fast Fourier transform (FFT)-based algorithms are used for range-Doppler detection. The detailed detection procedure is described below.

Figure 2 presents our design of the target detection and human–vehicle classification scheme based on 2D range-Doppler FMCW radar. In the fast-ramp-based FMCW radar, the received signal is sampled as $S_{a}\left(l,k\right)$ in the analog-to-digital converter (ADC) with sample rate $f_{s}$, where l=1−L is the sample number for every ramp and k=1−K is the ramp number during one transmit period.

If the single moving target and clutter are located in the FOV (field of view), the received radar signal is expressed by Equation (1), where $S_{t}\left(l,k\right)$ and $S_{c}\left(l,k\right)$ denote the signals reflected from the moving target and the clutter. Here, we assume that the moving target has Doppler scattering points of Q consisting of the Doppler frequency $f_{D}\left(q\right)$ and the corresponding amplitude $A_{t}\left(q\right)$, and all scattering points of the moving target are located in the same range. Generally, while there are multiple scattering points for a human, there are few for a vehicle. This assumption is described in the first term of Equation (1).

In this example, we also assume that one instance of clutter of the echo amplitude $A_{C}$ with zero-Doppler exists. In such a case, we can describe the received signal in detail, in this case using the second term of Equation (1), where $f_{r,t}$ and $f_{r,c}$ are correspondingly the range frequencies of the moving target and the clutter.
(1)$$S_{a}\left(l,k\right)=S_{t}\left(l,k\right)+S_{c}\left(l,k\right)=\;\left[\sum_{q=1}^{Q}A_{t}\left(q\right)\cdot e^{j2\pi \left(l-1\right)f_{r,t}}e^{j2\pi cT\left(k-1\right)f_{D\left(q\right)}}\right]+\left[A_{c}e^{j2\pi \left(l-1\right)f_{r,c}}\right]$$

Firstly, in the range-processing step, we extract the range-frequency spectra $Y_{a}\left(m,k\right)$ of every ramp by applying the windowing function $W_{R}\left(l\right)$ and M-point FFT in the received signal $S_{a}\left(l,k\right)$, where m=1−M is the range-bin number and k=1−K is the ramp number during a single transmit period. Equation (2) presents the range-processing results, where the first term $Y_{t}\left(m,k\right)$ indicates the complex spectrum of the moving target and the second term $Y_{c}\left(m,k\right)$ is that of the clutter.
(2)$$Y_{a}\left(m,k\right)=Y_{t}\left(m,k\right)+Y_{c}\left(m,k\right)=\;\overset{L}{\underset{l=1}{\sum }}S_{t}\left(l,k\right)W_{R}\left(l\right)e^{-j2\pi lm/M}+\overset{L}{\underset{l=1}{\sum }}S_{c}\left(l,k\right)W_{R}\left(l\right)e^{-j2\pi lm/M}$$

In order to suppress clutter, we employ an earlier study. In the clutter-suppression part, the components without a phase change in all ramps are removed using coherent subtraction and coherent summation [22]. In other words, in this step, clutter with near zero-Doppler characteristics can be suppressed and the moving target with the radial velocity can survive.

Thus, because we assume that the clutter has exactly zero-Doppler characteristics, the output of this step can be $Y_{t}\left(m,k\right)$. This indicates that the term of the moving target survives in Equation (2).

In the Doppler-processing step, the range-spectrum $Y_{t}\left(m,k\right)$ of the range-time domain is transferred by the windowing function $W_{D}\left(k\right)$ and N-point FFT in order to extract the range-Doppler spectrum $Z_{t}\left(m,n\right)$, expressed as Equation (3). Here, m=1−M is a range-bin number and n=1−N is the Doppler-bin number. In Figure 2, $X_{t}\left(m,n\right)$ is saved into 2D range-Doppler memory.
(3)$$Z_{t}\left(m,n\right)\approx \overset{K}{\underset{k=1}{\sum }}Y_{t}\left(m,k\right)W_{D}\left(k\right)e^{-j2\pi kn/N}$$

Finally, the target decision step is carried out to determine whether or not each cell $Z_{t}\left(m,n\right)$ of the 2D range-Doppler is a target. Typically, for an adaptive thresholding method, a conventional CA-CFAR (cell-averaging constant false alarm rate) detector is used in the Doppler direction [23]. That is, when comparing the magnitude of every cell and the average magnitude of the corresponding neighboring cells, the range-bin index $n_{t}=1-N$ and Doppler-bin index $m_{t}=1-M$ are extracted.

If one moving target has multiple scattering points, multiple range-bins and Doppler-bins can be detected. In that case, the clustering algorithm should be added to group multiple scattering points as one point, generating the representative range-bin index $m_{t}$ and Doppler-bin index $n_{t}$. In this paper, we assume that only one range-bin and one Doppler-bin with the dominant magnitude exist.

In a conventional 2D range-Doppler FMCW radar system, the radar senor output is the target detection information, including the ranges and velocities of the detected targets. However, in this paper, the Doppler spectrum $D_{t}\left(n\right)$ of the detected target is also extracted in addition of the range-bin index $m_{t}$ and Doppler-bin index $n_{t}$ from the detection block, and all of this information is fed into the classification block, shown in Figure 2. Here, $D_{t}\left(n\right)$ denotes the entire Doppler spectrum placed in range-bin index $m_{t}$, and it is expressed as the absolute value of $\left\{Z_{t}\left(m_{t},n\right),\;n=1-N\right\}$.

Based on the detection information, we propose a feature-based human–vehicle classification scheme using machine learning for an FMCW radar sensor, as presented in Figure 3.

In the proposed human–vehicle classification algorithm, the two steps are divided. That is, the “scattering-point feature extraction” step and “magnitude feature extraction” step generate three features ($x_{1}$, $x_{2}$, and $x_{3}$) using the output of the detection part. These three extracted features are fed into the machine-learning engine for learning and testing.

Firstly, in the “scattering-point feature extraction” step shown in the upper left part of Figure 3, we extract two features, $x_{1}$ and $x_{2}$ to represent the characteristics of the Doppler spectrum distribution in the frequency domain, representing the extent of the Doppler expansion and the degree of change of the Doppler spectra.

In order to measure the extent of the Doppler spectrum expansion in the current frame, we count the Doppler reflection points of the detected target with power exceeding the reference threshold $T_{mag}$. That is, the first feature $x_{1}$ is calculated by counting the number of surviving points among the scattering points of the Doppler spectrum of target. The magnitude threshold $T_{mag}$ was found to have a value 10 dB less than maximum magnitude $D_{t}\left(n_{t}\right)$ in an earlier study [19].

The $x_{1}$ feature can be implemented via the summation of the logical output to meet the requirement of $D_{t}\left(n\right)>T_{mag}$. The mathematical expression is presented here as Equation (4), where $\hat{D_{t}}\left(n\right)$ is expressed as only 1 or 0.

If the detected target is a human, the $x_{1}$ feature will be much higher than that in the vehicle detection case because various Doppler features are reflected from human components such as body, limbs, and arms. Thus, in this paper, we newly define the $x_{1}$ feature as SPC (scattering point count). Here, $S_{t}$ ($=x_{1}$) is always maintained kept through one delay element as $S_{t}^{\prime }$ to generate the second feature $x_{2}$ for the processing of next frame.
(4)$$x_{1}=S_{t}=\sum_{n=1}^{N}\hat{D_{t}}\left(n\right),\;\mathrm{where}\;\hat{D_{t}}\left(n\right)=\left\{\begin{array}{cc} if\;\hat{D_{t}}\left(n\right)>T_{mag}, & 1\\ otherwhise, & 0 \end{array}\right\}$$

Secondly, in order to measure the time variance of Doppler spectrum shape, we calculate the absolute distance in the number of Doppler reflection points between two successive frames. We newly define the feature as the SPD (scattering point difference), defined by Equation (5). That is, this feature indicates the degree of change of the Doppler spectra.

Thus, this $x_{2}$ feature of a walking human will much higher than that of a vehicle in many cases, because the shapes of the Doppler spectra of a vehicle are similar regardless of time, whereas the echo spectrum shape of a human changes over time.
(5)$$x_{2}=\left|S_{t}-S_{t}^{\prime }\right|$$

Finally, in order to consider that the echo power of a walking human can vary due to magnitude fluctuations of the received signal, we propose a third feature, $x_{3}$. The extraction of this feature is carried out during the “magnitude feature extraction” step in the right part of Figure 3.

Using the detected Doppler-bin index $n_{t}$ and the corresponding Doppler spectrum $D_{t}\left(n\right)$, we can calculate the maximum power of the indicated $n_{t}$-th Doppler-bin as $P_{t}=D_{t}\left(n_{t}\right)$.

The power reflected from the target depends on the RCS of target, and the RCS of a human generally is much lower than that of a vehicle. However, it is nearly impossible to determine the class of object using only the echo power, as the received reflections can vary according to the shape of the object, as well as the target position, the moving direction of the object, and various target conditions.

Thus, in this paper, we define the difference rate between the reflected echoes in two successive frames as the MDR (magnitude difference rate).

However, in the last frame and the current frame, despite the fact that each magnitude $P_{t}$ is detected from the same target, two magnitudes can be reflected from a different range. That is, it is necessary to consider that the target can move over two successive measurement times. Thus, we should compensate for path loss in the extracted maximum power $P_{t}$ in order to generate the MDR. Thus, we calculate $G_{t}$ as the magnitude normalized by the distance to the power of 4. Here, $G_{t}$ is always stored for this processing of next frame, and the absolute difference rate $x_{3}$ is ascertained using $G_{t}$ and $G_{t}^{\prime }$ as shown in Equation (6). Here, $G_{t}^{\prime }$ is the value in this processing output of the previous frame.
(6)$$x_{3}=\frac{\left|G_{t}-G_{t}^{\prime }\right|}{G_{t}},\;\mathrm{where}\;G_{t}=\frac{P_{t}}{{\left(R_{t}\right)}^{4}}=\frac{D_{t}\left(n_{t}\right)}{{\left(R_{t}\right)}^{4}}$$

In this case, because we need the range value $R_{t}$ of the target located in the $m_{t}$-th range-bin, we calculate $R_{t}$ using the detected range-bin index $m_{t}$ with Equation (7) in advance.
(7)$$R_{t}=\frac{cf_{r,t}}{2BT}=\frac{c\left(1-m_{t}\right)\left(\Delta f\right)}{2BT}=\frac{c\left(1-n_{t}\right)\left(f_{s}/M\right)}{2BT}$$

As described thus far in this paper, the concept of the proposed Doppler spectrum features is illustrated in Figure 4.

We apply machine learning in order to classify a human and a vehicle using the three extracted features. In this paper, for machine learning, we employ the SVM (support vector machine) and BDT (binary decision tree) methods.

An SVM is a popular and simple machine learning algorithm, and it is a bisection method that determines the best classifier, which divides the given data into two difference groups. Thus, SVM is broadly used for target classification in radar signal processing [24].

Moreover, BDT is also a simple structure based on a sequential decision process because a feature is evaluated as one of two branches, which is selected starting from the root of tree [25].

Thus, if the performances of SVM and BDT are similar, we can easily implement both SVM and BDT into an embedded system for machine learning based on the three features proposed in this paper, using only “if–else” syntax in real time.

## 3. Measurement Results

In order to verify the performance of the proposed human–vehicle classification scheme based on Doppler spectrum features, we set up the measurement environment using a radar front-end module and a real-time data acquisition module, as shown in Figure 5.

We employ a radar front-end module consisting of a 24-GHz FMCW transceiver, used in previous work [26], and a newly designed radar patch antenna with an FOV of 80°. The transceiver and antenna are prototypes developed by DGIST for academic research and experiments. Thus, in the radar sensor, we can change certain radar modulation parameters according to the designed system budget to meet the requirements of specific applications.

The received radar signal is digitalized and transferred to a personal computer (PC) through an Ethernet connection. The designed hardware specifications and system parameters are shown in Table 1.

In this paper, we consider five scenarios in an outdoor environment, as shown in Figure 6. Photos of a walking human and a moving vehicle for each scenario are presented in Figure 7. Detailed descriptions are given in Table 2.

In this paper, there were 1520 whole frames for actual objects. Among the whole frames, there were 1080 human detections and 460 vehicle detections, respectively.

Figure 8 shows the range-Doppler maps of five successive frames for scenario #1 for a walking human and scenario #5 for a moving vehicle. Here, the *x*-axis is the velocity (m/s) and the *y*-axis indicates the range (m). While it can be observed that the Doppler spectra of a moving vehicle are similar, given the sharp shape shown in Figure 8a, those of the walking human fluctuate due to the non-rigid motion, as shown in Figure 8b.

At some looking angles, the motion of the wheels of a vehicle can cause broad spreading of the Doppler spectrum. In a related study [12], when the wheels of a vehicle were directly visible within about 45° from the location of the radar sensor, the micro-Doppler signal could be detected. Thus, this can reduce the decision performance of the proposed human–vehicle classification scheme.

However, the micro-Doppler component from the wheel is quite small in most cases compared to the overall gross motion of the Doppler form [27]. Moreover, if using a radar sensor with multiple receive channels, we can use the angular spectrum based on DBF (digital beam forming). When the side of the vehicle is reflected, the angular spectrum can also be extended. We will enhance the human–vehicle algorithm based on these characteristics in the future.

Figure 9 shows the range profile of the moving vehicle in scenario #5 and the walking human in scenario #1. In the range-profiles, the *x*-axis is the frame index, and the *y*-axis indicates the range (m).

According to measurement results using the radar prototype used in this paper, we found that the maximum detectable range is approximately 14 m. Thus, we extracted the Doppler spectra at about 21 m and 13 m from the detection results of the vehicle and human, respectively. The right side of Figure 9 shows these Doppler spectra of two successive frames for both targets. Here, the *x*-axis is the velocity (m/s), and the y-axis indicates the magnitude normalized by the range.

Even at the longer range, we can find that the Doppler spectra of a human can still be extended and can vary compared to those of a vehicle. That is, if a human and a vehicle can be detected, the Doppler spectrum characteristics proposed in this paper are shown regardless of the detected range.

Figure 10a–c correspondingly present the two-dimensional distributions of $x_{1}$–$x_{2}$, $x_{2}$–$x_{3}$, and $x_{3}$–$x_{1}$. Here, the red stars and blue circles represent the actual classes of a moving vehicle and a walking human, respectively.

In Figure 10a, we find that the red stars are mostly positioned in the lower left area. On the other hand, the blue circles are mostly scattered out of the area where the red stars are gathered. These results indicate that the Doppler spectrum extension feature and Doppler spectrum variation feature for the vehicle are low compared to those of the walking human. Therefore, the SPC and SPD features are very useful to identify a walking human and a moving vehicle.

In Figure 10b,c, we find that the distributions of the third feature are mostly gathered in narrow areas, with some spread over the entire area. In a more detailed analysis, we find that the MDR features for the human are somewhat more widely distributed than those of the vehicle. However, these results cannot determine between two classes using only the values of the magnitude difference rate between the reflected echoes in two successive frames.

Therefore, in order to distinguish between a walking human and a moving vehicle, it is necessary to use the MDR feature together with the SPC and SPD features with machine learning.

The training and test process to verify the performance of the proposed human–vehicle classification is presented in Figure 11. In this paper, the procedure describing all programming for machine learning and verification was as follows:We separated the three feature vectors of the actual human randomly, with 80% for training and 20% for testing. In the same way, the data for an actual vehicle were also randomly divided such that the learning set and the validation set were 80% and 20%, respectively. In advance, we labeled the actual human as “8” and the actual vehicle as “1” into feature vectors consisting of the three features. These feature vectors are illustrated in the left part of Figure 10, where the gray and white areas indicate the data for the actual human and those for the actual vehicle, respectively.We employed the “*fitcsvm (data, label, options)*” and “*fitctree (data, label, options)*” functions provided in Matlab to code the proposed algorithms, where both functions returned a trained SVM and a fitted BDT based on the input variables comprising the data and labels.We also optimized the SVM and BDS parameters via a 30-trial loop with the training set.We input the test dataset into the completed SVM and BDT engines to verify the performance of the proposed algorithm.For an effective verification, we repeated the entire procedure 10 times, including the separation of the dataset, the training, and the validation steps.We could check the performance of the proposed human–vehicle classification algorithm by averaging the results of the 10 aforementioned processes.

To summarize all of the procedures described thus far, we describe the classification processing steps in Figure 12.

Figure 13a shows one part of the optimized SVM with three training data instances labeled for the vehicle and human classes, where the results are in the $x_{1}$–$x_{2}$–$x_{3}$ domain. The black squares are the support vectors, and the blue squares and red stars correspondingly indicate the features of the vehicle and the human.

Figure 13b graphically presents the final optimization results of the BDT. The BDT was structured with conditional nodes using the three features. In the final nodes, “8” and “1” indicate the vehicle and human classes, respectively.

To compare the performance capabilities during the human–vehicle classification task, we define the four classification algorithms combining the three features, as presented in Table 3.

Firstly, in typical algorithm #1, we simply use the magnitude reflected echoes in the current frame for classification. To do this, we newly define the MCV (magnitude current value), expressed as $x_{3}=G_{t}$, which is normalized by the detected range.

Secondly, typical algorithm #2 is a method that uses only the SPC feature, which indicates the extent of the Doppler spectrum.

Next, in the proposed algorithm #1, the SPD feature is added, meaning that we consider the variation of the extent of the Doppler spectrum together with the first method. Finally, the proposed algorithm #2 also includes the MDR feature, meaning that the variance of the Doppler echo is also considered.

Thus, in Table 4 and Table 5, we present the classification decision rate as the processing results of the designed machine learning for the typical algorithms #1 and #2, and the proposed algorithms #1 and #2. In these matrices, the horizontal axis indicates the actual classes of the target, and the vertical axis indicates the classification algorithms used.

In Table 4, which contains the SVM results, the results of typical algorithm #1 indicate that the method using only the reflected magnitude is not suitable as a human–vehicle classification method. In contrast, the results of typical algorithm #2 with only ($x_{1}$) show that the classification decision rates for a human and a vehicle were 90.87% and 91.78%, respectively.

However, with the proposed algorithm #1 with two features ($x_{1}$ and $x_{2}$), the corresponding performances were estimated to be 96.58% and 91.90%. In this case, while the result for the walking human increased by 5.71%, we found only a slight increase in the decision rate for the moving vehicle. This indicates that a Doppler spectrum extension or some variations are occasionally found, even when the target is a vehicle.

In addition, in the proposed algorithm #2, using three features ($x_{1}$, $x_{2}$, and $x_{3}$), we find that the classification decision rates for the human and vehicle correspondingly increased by 8.13% and 4.65%. As compared to those results obtained using the typical algorithm #2, this means that the radar Doppler frequency spectrum for a walking human has more fluctuation of the magnitude and more variation in the width compared to the vehicle case.

Next, Table 5 presents the classification decision rate using BDT machine learning. As compared to the SVM outcomes, we find that these results were similar or slightly better in the proposed algorithms #1 and #2. Moreover, the BDT-based classification algorithm is slightly more useful for human recognition than for vehicles.

Because the proposed algorithm uses three features extracted from only two successive frames, real-time implementation for an embedded system is more feasible. Moreover, the structures of the SVM and BDT can be implemented simply using the “if–else” syntax. Therefore, we think that the proposed Doppler-spectrum feature-based human–vehicle classification scheme using SVM or BDT is useful as a type of embedded software design.

## 4. Conclusions

In this paper, we proposed a human–vehicle classification scheme using a Doppler-spectrum feature-based SVM and BDT for a commercial FMCW radar system. We define three new features, referred to as SPC, SPD, and MDR, to express the reflected the characteristics of the Doppler spectrum.

We extracted three features from a received radar signal measured by a 24-GHz FMCW radar front-end module and a real-time data acquisition module. The features were used as input data for the SVM and BDT. Then, through 10 randomly repeated verification trials, we found that the classification decision rate for an actual human and a vehicle exceeded 99% and 96%, respectively, for both machine learning engines. These results demonstrate that the performance outcomes using the proposed the three features were approximately 8% and 5% higher for a human and a vehicle compared to those when using the only first feature.

Because the Doppler spectra in two successive frames were only used and the procedure to extract the features was very simple, the proposed method can easily be implemented in real time. In addition, because the software structures of the BDT and the SVM were simple, the proposed human–vehicle classification scheme would be useful for real-time embedded implementations.

In the future, we plan to apply the proposed classification scheme to a multiple-target situation. Specifically, we will consider a traffic situation in which multiple vehicles can appear at the same time, with the possible spreading of Doppler spectrum. We will also use various types of targets, for example, motorcycles, bicycles, E-scooters, and personal mobility devices, with a human standing or sitting on these targets, as well as a stroller with a human pushing or pulling it.

To do this, we plan to employ a radar front-end module with multiple receive antennas in order to estimate the angular spectrum of the target. Thus, we will extract various features based on the range-Doppler angular spectrum and design a multi-class and multi-target classification scheme capable of real-time processing based on the method proposed in this paper.

## Figures and Tables

**Figure 1 sensors-20-02001-f001:**
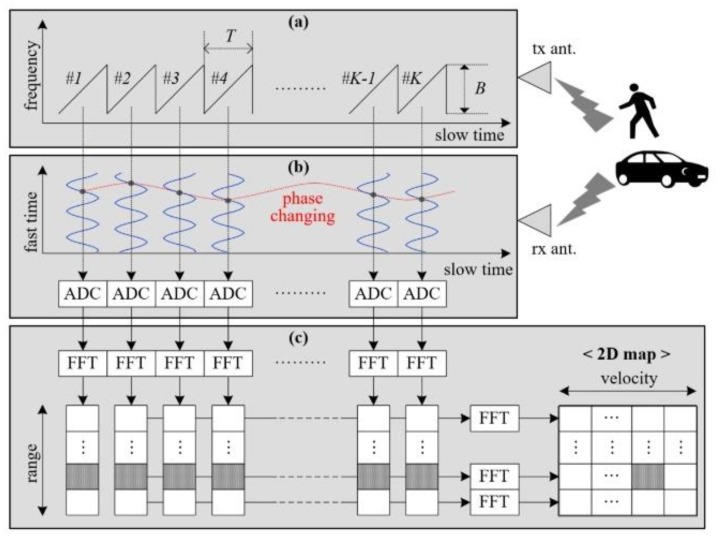
Basic concept of fast-ramp-based two-dimensional (2D) range-Doppler frequency-modulated continuous wave (FMCW) radar. Here, T is the modulation period, B is the bandwidth, and K is the number of ramps: (**a**) transmitted signal in the frequency and slow-time domain; (**b**) received beat signal for a single moving target in the fast-time and slow-time domains; (**c**) signal-processing scheme for 2D range-Doppler map generation using two-step fast Fourier transform (FFT).

**Figure 2 sensors-20-02001-f002:**
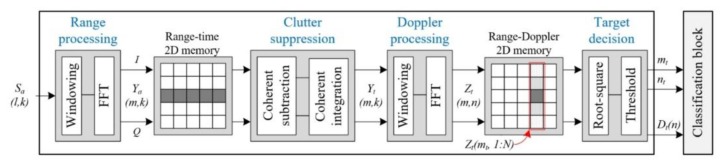
Proposed radar signal-processing method using a target detection block and a classification block.

**Figure 3 sensors-20-02001-f003:**
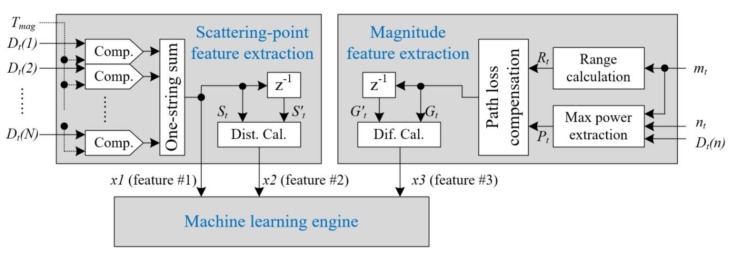
Proposed human and vehicle classification algorithm using the feature extraction and machine learning processes.

**Figure 4 sensors-20-02001-f004:**
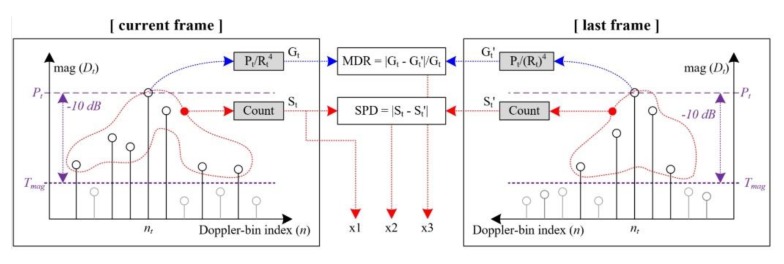
Concept of the proposed features using the Doppler spectra of the same target in the last and current frames.

**Figure 5 sensors-20-02001-f005:**
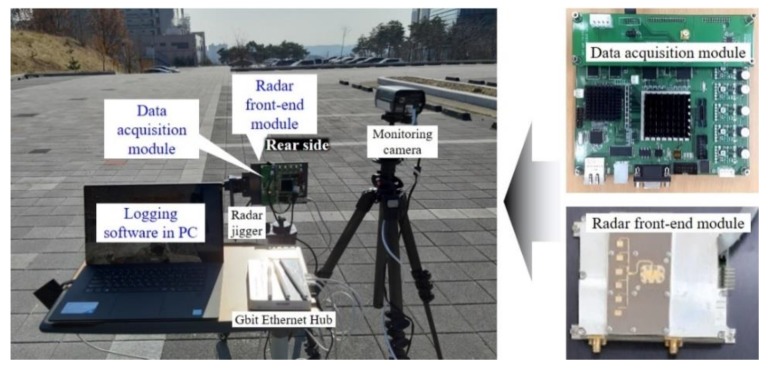
Photo of the measurement set-up using the 24-GHz FMCW front-end module and real-time data acquisition module.

**Figure 6 sensors-20-02001-f006:**
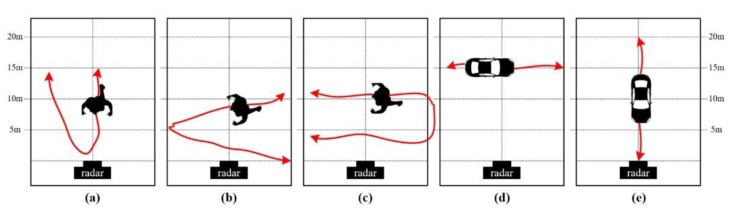
Configurations of the five measurement scenarios in an outdoor environment: (**a**) scenario #1; (**b**) scenario #2; (**c**) scenario #3; (**d**) scenario #4; (**e**) scenario #5.

**Figure 7 sensors-20-02001-f007:**
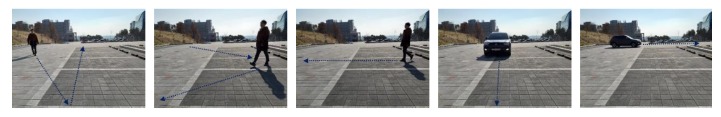
Photos of scenarios #1–#5.

**Figure 8 sensors-20-02001-f008:**
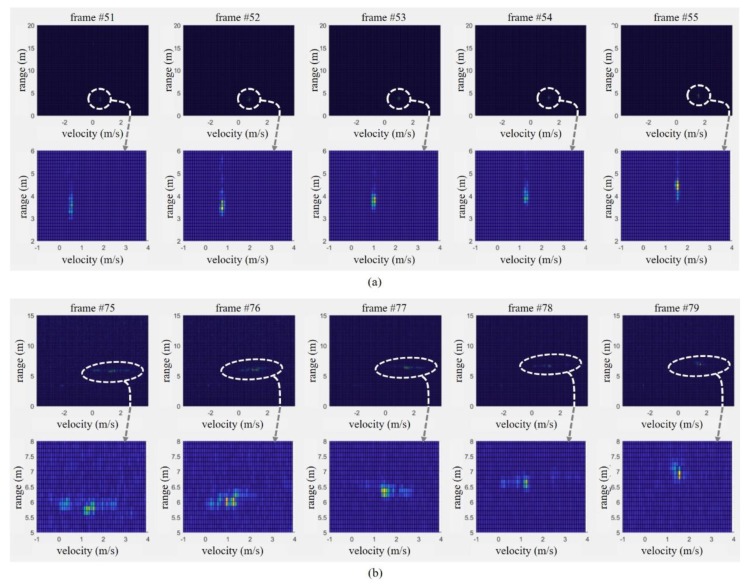
Range-Doppler map of the measurement results using the proposed detection algorithm: (**a**) scenario #5 for a moving vehicle; (**b**) scenario #1 for a walking human.

**Figure 9 sensors-20-02001-f009:**
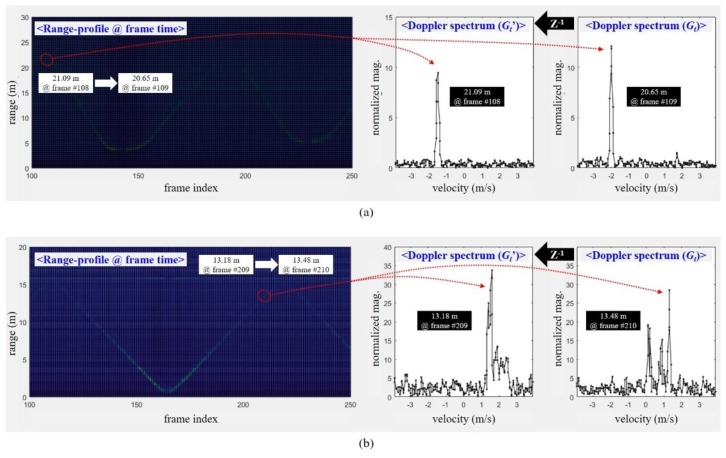
Detected range-profile in frame time domain and the corresponding Doppler spectra of two specific successive frames for feature extraction: (**a**) scenario #5 for a moving vehicle; (**b**) scenario #1 for a walking human.

**Figure 10 sensors-20-02001-f010:**
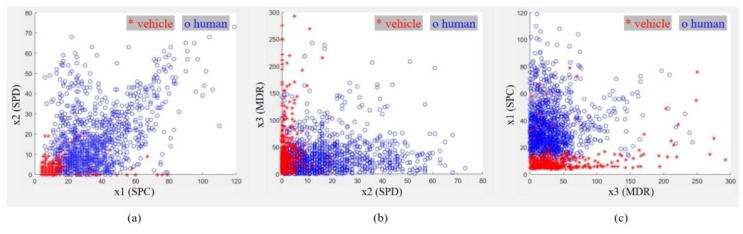
Two-dimensional distribution of three features for humans and vehicles: (**a**) results of the $x_{1}$–$x_{2}$ features; (**b**) results of the $x_{2}$ –$x_{3}$ features; (**c**) results of the $x_{3}$–$x_{1}$ features. Here, the red stars denote the features of a vehicle and the blue circles indicate those of a human.

**Figure 11 sensors-20-02001-f011:**
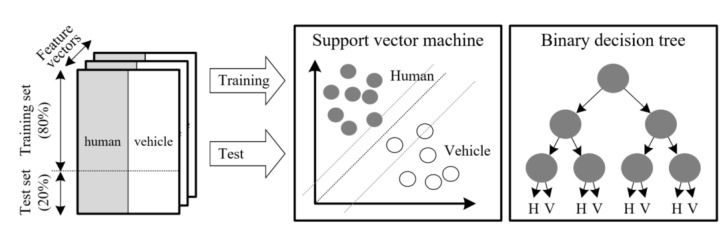
Structure of features comprising the training dataset and test set.

**Figure 12 sensors-20-02001-f012:**
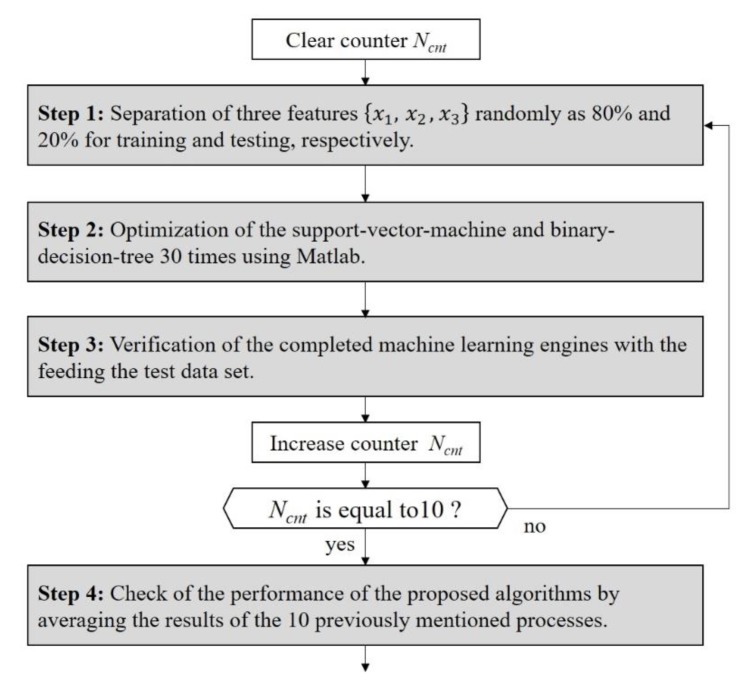
The training and test procedure using machine learning with the extracted features.

**Figure 13 sensors-20-02001-f013:**
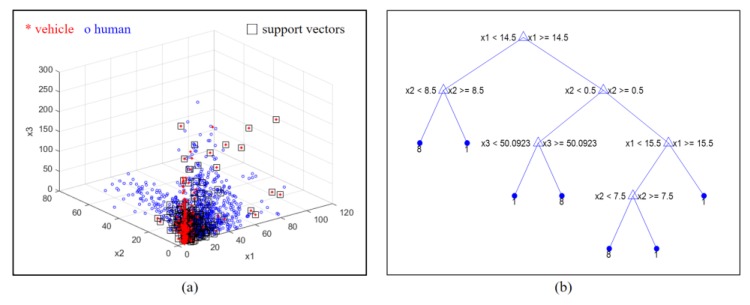
Examples of both machine learning engines optimized by Matlab using the proposed feature based on the Doppler spectrum: (**a**) of results of support vector machine; (**b**) results of binary decision tree.

**Table 1 sensors-20-02001-t001:** Parameters of the radar system used in this paper. FOV—field of view; ADC—analog-to-digital converter; SDRAM—synchronous dynamic random-access memory; FFT—fast Fourier transform.

Parts	Specifications	Units	Symbols	Values
Front-end module	Center frequency	GHz	f_c_	24
	Bandwidth	GHz	B	1
	Number of ramps	-	K	256
	Number of antennas	-	-	1
	Modulation period	μs	T	400
	FOV	-	°	80
Logging Board	ADC sampling rate	MHz	f_s_	5
	One frame time	ms	-	200
	SDRAM size	MB	-	256
	Ethernet	Mb/s	-	1000
FFT processing	Range-FFT point	-	M	2048
	Doppler-FFT point	-	N	256

**Table 2 sensors-20-02001-t002:** Measurement scenario descriptions.

Scenarios	Descriptions
1	A human is moving along the middle line of the radar sensor or some degrees away from the centerline within approximately 15 m.
2	A human is crossing the road diagonally at a 5-m interval within approximately 15 m.
3	A human is laterally moving 10 m or 5 m away parallel to the radar system.
4	A vehicle is moving along the middle line of the radar sensor within approximately 20 m.
5	A vehicle is laterally moving 15 m away parallel to the radar system.

**Table 3 sensors-20-02001-t003:** Four classification algorithms used with a combination of three features. MCV—magnitude current value; SPC—scattering point count; SPD—scattering point difference; MDR—magnitude difference rate.

	Features	*x* _1_	*x* _2_	*x* _3_
Algorithms				
typical algorithm #1		-	-	MCV
typical algorithm #2		SPC	-	-
proposed algorithm #1		SPC	SPD	-
proposed algorithm #2		SPC	SPD	MDR

**Table 4 sensors-20-02001-t004:** Classification decision rate based on the SVM (support vector machine).

	Actual Class	Human	Vehicle
Algorithms			
typical algorithm #1		95.89%	8.39%
typical algorithm #2		90.87%	91.78%
proposed algorithm #1		96.58%	91.90%
proposed algorithm #2		99.00%	96.43%

**Table 5 sensors-20-02001-t005:** Classification decision rate based on the BDT (binary decision tree).

	Actual Class	Human	Vehicle
Algorithms			
typical algorithm #1		95.90%	5.32%
typical algorithm #2		90.83%	90.88%
proposed algorithm #1		97.09%	92.41%
proposed algorithm #2		99.27%	96.70%

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
