# Peer review of "Doppler-Spectrum Feature-Based Human–Vehicle Classification Scheme Using Machine Learning for an FMCW Radar Sensor"

_sensors, 2020, doi:10.3390/s20072001_

Round 1

Reviewer 1 Report

Abstract:

The abstract is clear, just a few comments on grammar:

Line 12: for an FMCW (...) radar sensor

Lines 12-13: We introduce three novel features

Line 20: for the walking human

Introduction:

Please refer to your paper as "This paper" or "This work" (e.g.) instead of "This research"

I would prefer a high-level motivation in the Introduction section and then split out your literature review into a related work section.

Some minor fixes:

Line 63: are limited

Line 83: can fluctuate

Line 85: We have previously presented...

Line 92: delete machine.

Proposed Human Indication Scheme

Can you make it clear if you are using your own custom hardware or an off-the-shelf sensor that allows you to change the radar modulation etc?

Generally, I feel like there is a lot of math that isn't motivated by intuition/explanation

Some minor comments:

Line 112: algorithms are used

Line 113: the detailed

Line 116: split this set/equation onto a new line

Line 136: split this set/equation onto a new line

Line 140: We employ an earlier study

Line 149: split this set/equation onto a new line

Can you split the content of this section into subsections, perhaps "feature engineering" and "training/learning"

Measurement Results

The results section is significantly better written, thank you.

Some minor comments:

Line 270: respectively

Line 278: While it can be observed

Line 303: I don't really care that it was coded in Matlab, unless the code is open-sourced.

Similarly Line 313: describe what fitcsvm and fitctree are doing (reference to your method Sec 2) instead of listing the Matlab functions

Lines 310-324 are quite badly page-formatted (lots of paragraphs) - would it be better to have a bullet or numbered list?

Results seem clearly better than a typical algorithm, congratulations.

Author Response

We appreciate the reviewers for their great efforts to review our paper.

We also would like to acknowledge the constructive comments made by the reviewers.
First, we have provided a reply for each comment, such as the attached "reply letter".

We have also revised the manuscript by incorporating these suggestions.
Moreover, all additions and corrections are marked in the revised paper.

Reviewer 2 Report

The approach (classification of FMCW data) is not new, relevant references are missing.

The measurement results (Doppler signature) do not support the statements in the text.

The approach (proposed features) is not compared to other solutions.

The research design with only two classes is not challenging.

Author Response

(The authors gave the same response as above.)

Reviewer 3 Report

This paper addresses the classification of human and vehicle using two successive Doppler spectrums obtained by FMCW radar. The three features exploited for classification are well explained and the whole processing procedure can be easily understood.

Just a few questions/suggestions here.

(1) Feature x1: this feature is based on the observation that the Doppler spectrum spreading of a human can be boarder than that of vehicle because the human has the non-rigid micro-motion caused by the arms and legs, etc.

I agree with this in most cases, but I’m still curious for 3 more scenarios:

  • Very near range case:

Because at the nearer range or in some looking angles, the motion of the wheel of the vehicle can also cause a wide spreading of the Doppler spectrum, and affect the discrimination capability of feature x1. I am curious to what extent, the wheel of the vehicle can cause an issue in classification.

  • Long range cases:

At long ranges, the spreading of the human Doppler spectrum will be less due to the weak returns of the micro-motion, and might affect the discrimination between the human and vehicle. If the experiment allows, it would be better to include some long-range analysis.

  • Traffic junction

At traffic junction, multiple vehicles may appear at the same time and may cause the spreading in Doppler axis, and further influence the discrimination. It would be very interesting to investigate such case for the future work.

(2) Feature x2: this feature is based on the observation that the variation of human Doppler spectrum in two successive time frames is greater than that of vehicle.

Similar to feature x1, it would be better to include long range analysis. Because I guess this feature is also related to range. For example, at near range, the detectable variation of human motion can be large, but for the long range, the variation may not be entirely detectable and noticeable by the radar. The interesting is how the range affects the classification performance.

(3) A minor error

In line 160, 168, it is defined that the range index is nt and Doppler index is mt. But in line 170, perhaps there is a mistake for mt.

I understand this paper focuses on the classification between the human (mainly on walking person in this paper) and vehicle. In recent years, due to the boom of online ordering and the food delivery, there is the massive emergence of the E-scooter and personal mobility devices (PMD) with the human standing or sitting on it. It is relatively more difficult to discriminate the E-scooter or PMD from the vehicle as the human in this case almost has no micro-motion. But it is very interesting to see some analysis for this in the future and the demand is really there.

Author Response

(The authors gave the same response as above.)

Round 2

Reviewer 2 Report

The claim of the authors, that the proposed approach can be extended to more realistic and thus more complex szenarios is very much doubtable.

The claim of the authors, that their approach has a significant advantage for implementation on embedded systems is very much doubtable (in order to obtain a range doppler map, on which the approach relies, a real time 2D FFT is necessary).

The number of data used for each szenario (Table 2) is missing.

The decission rate of the solution for each szenario is missing

The MATLAB figures should not be imported as screen shots.
